# Genomic Regions Associated with the Position and Number of Hair Whorls in Horses

**DOI:** 10.3390/ani11102925

**Published:** 2021-10-10

**Authors:** Diogo Felipe Pereira de Assis Lima, Valdecy Aparecida Rocha da Cruz, Guilherme Luís Pereira, Rogério Abdallah Curi, Raphael Bermal Costa, Gregório Miguel Ferreira de Camargo

**Affiliations:** 1Escola de Medicina Veterinária e Zootecnia, Universidade Federal da Bahia (UFBA), Salvador 40170-110, BA, Brazil; dflima@live.com (D.F.P.d.A.L.); valdecya.r.cruz@gmail.com (V.A.R.d.C.); raphaelbcosta@gmail.com (R.B.C.); 2Departamento de Melhoramento e Nutrição Animal, Universidade Estadual Paulista (Unesp), Botucatu 18618-681, SP, Brazil; guilherme.luis@unesp.br (G.L.P.); rogerio.curi@unesp.br (R.A.C.)

**Keywords:** behavior, hair follicle, Quarter Horse, SNP, temperament

## Abstract

**Simple Summary:**

Whorls have been used to indicate the temperaments of domestic animals; however, little is known about the biological events that drive this association. The present study is the first that aims to find the main genomic regions that influence the whorl traits in livestock, with horses as a model. Genes related to hair follicle growth were found. Interestingly, some of these genes also influence psychiatric diseases and neurological disorders, thus evidencing a consistent biological explanation for the association.

**Abstract:**

The position and number of hair whorls have been associated with the behavior, temperament, and laterality of horses. The easy observation of whorls assists in the prediction of reactivity, and thus permits the development of better measures of handling, training, mounting, and riding horses. However, little is known about the genetics involved in the formation of hair whorls. Therefore, the aim of this study was to perform a genome-wide association analysis to identify chromosome regions and candidate genes associated with hair whorl traits. Data from 342 Quarter Horses genotyped for approximately 53,000 SNPs were used in an association study using a single-step procedure. The following traits were analyzed: vertical position of hair whorl on the head, number of whorls on the head, and number of whorls on the left and right sides of the neck. The traits had between one and three genomic windows associated. Each of them explained at least 4% of the additive variance. The windows accounted for 20–80% of additive variance for each trait analyzed. Many of the prospected genes are related to hair follicle growth. Some of these genes exert a pleiotropic effect on neurological and behavioral traits. This is the first indication of biological and physiological activity that might explain the association of hair whorls and temperament.

## 1. Introduction

Quarter Horses have great visibility and economic importance, mainly due to their versatility in different equestrian modalities. The breed is subdivided into different populations according to their competencies, resulting from different selection objectives. The racing population has animals with the ability to run short distances, while the stock population is used in functional tests, exploring skills such as agility, temperament, obedience, and cow sense [1].

Horses are animals that are managed directly and routinely by humans. Behavioral differences are easily observed in this species [2]. However, the measurement of temperament traits is not always easy. A morphological mark that has been associated with temperament is the hair whorl. In horses, the vertical and lateral position, number, and direction of the growth of hair whorls on the head have been associated with behavior, temperament, and laterality [3,4,5,6]. This association has been attributed to the fact that the epidermis and nervous system have the same embryonic origin [7]. 

Hair whorl traits in horses show high heritability [8,9,10] and are highly correlated with one another [10]. However, the genetic architecture of these traits and the biological mechanisms underlying their association with animal behavior are unknown. Therefore, the aim of this study is to perform the first genomic-wide association study (GWAS) of the vertical position of hair whorls on the head, in addition to the number of whorls on the head and neck in domestic animals, using horses as a study model. It will help to construct the first genetic–biological explanation of the association between whorls and temperament.

## 2. Materials and Methods

This study used 340 registered Quarter Horses of the racing line, 265 females and 75 males, born between 1985 and 2012. For the evaluation of whorl position, animals with hair whorls in more than one vertical position on the head were excluded, resulting in 252 females and 73 males. 

The total sample was established to best represent the diversity of racing Quarter Horses in Brazil. Animals with many offspring, as well as descendants of individuals that were influential in the formation of the breed, were selected to be genotyped. Sampling of full siblings was avoided. The breeders of these animals agreed to the sampling and genotyping. The animal procedures were approved by the Ethics Committee on Animal Use of FMVZ, Unesp, Botucatu (protocol number 157/2014–CEUA).

The hair whorl phenotypes were obtained from the official identification document (outline diagram) kindly provided by the Brazilian Association of Quarter Horse Breeders (ABQM), which contains graphical and discretionary representations of the location of the marks that identify the animals (Figure 1).

The following traits were studied: the position (POS) and number (NUM) of hair whorls on the head, and the number of hair whorls on the left (NUML) and right (NUMR) side of the neck. For POS, the horses were classified into three groups according to whorl position: (1) above the upper eye line; (2) between the upper and lower eye line; and (3) below the lower eye line. For NUM, the hair whorls arranged on the animal’s head were counted, considering the head area to end at the appearance of the ears. For NUML and NUMR, the number of hair whorls located throughout the neck was counted, considering the appearance of the cranium and thorax as the upper and lower limits, respectively. 

Preliminary analyses of data consistency and descriptive statistics were performed using the R package (https://www.r-project.org). Table 1 shows the descriptive statistics of the population. The systematic effects of sex and hair coat were tested in the model and were not significant (*p* > 0.05).

Classes with few observations for NUM, NUMRL, and NUMR were also grouped to other classes to avoid low-incidence sampling problems. The analyses were re-run, and the results proved to be the same. 

### Genome-Wide Association Study

A number of the animals (n = 120) were genotyped with the Illumina Equine SNP50 BeadChip (54K) (Illumina Inc., San Diego, CA, USA) and the rest (n = 240) with the Illumina Equine SNP70 BeadChip (65K) (Illumina Inc., USA). Quality control of genotyped individuals and SNPs and genotype imputation were performed as described by [11]. Animals with a call rate of < 0.9 were excluded from the data set. SNPs located on the X-chromosome, SNPs with a call rate of <0.9, and SNPs with a *p*-value of <1 × 10^−5^ for Hardy–Weinberg equilibrium were eliminated. The MAF was not initially used as an exclusion criterion of SNPs, since different ranges of MAF were applied to verify the efficiency of the two-step genotype imputation between the SNP50 and SNP70 equine chips. After quality control and imputation, 342 animals genotyped with 55,196 SNPs remained. The SNPs of the chip were annotated in the EquCab 2.0 genome assembly, and their position was updated to the most recent version of the genome assembly, EquCab 3.0. Some SNPs could not be aligned, and had to be excluded from the analysis, leaving 53,828 markers remaining.

Genomic association analysis was performed using four independent phenotypes: (1) POS; (2) NUM; (3) NUML; and (4) NUMR, in a single-trait model for each phenotype evaluated.

The threshold animal model without systematic effects assumes that the underlying (liability) scale has a normal continuous distribution as follows:$$l_{ij}=a_{i}+e_{ij},$$
where ***l*** is the liability; ***a*** is the random effect of the animal; and ***e_ij_*** is the residual random vector. The animal model using Bayesian inference assumes that $\sigma_{a}^{2}\sim N(0.A\;\sigma_{a}^{2})$ and $\sigma_{e}^{2}\sim N(0.R\;\sigma_{e}^{2})$, where $\sigma_{a}^{2}$ and $\sigma_{e}^{2}$ are the direct genetic and residual variance components, respectively; ***A*** is the numerator relationship matrix, and ***R*** is the variance matrix of the residual vector. The residual variance ($\sigma_{e}^{2}$) was set at 1. According to the Bayesian approach, liability is conditional on all parameters, and is thus independently distributed, allowing the estimation of mean breeding values different from zero. Convergence was evaluated by Geweke criteria that varied from 0.0 to 0.02 for the analyses.

Quality control of the SNPs after imputation was performed using PEGSF90 software [12] which considered a call rate of >0.90, a minor allele frequency of >0.05, and a Hardy–Weinberg equilibrium of <0.15. A total of 53,828 markers remained for GWAS. 

Single-step GBLUP was used for analysis. The variance components were estimated with the THRGIBBS1F90 software [12], posterior means were obtained with POSTGIBBSF90 [12], and the POSTGSF90 software was used for the prediction of SNP effects, and for the generation of Manhattan plots. 

The variations explained by windows comprising 100 adjacent SNPs were used to identify genome regions with a major effect on the traits evaluated. Major-effect genes present in the windows were prospected in the NCBI database (National Center for Biotechnology Information) using EquCab 3.0 as the reference genome, filtered by species, chromosome, and position of the window. 

## 3. Results

The present results of GWAS of the position and number of hair whorls on the head and neck of horses are the first reported in the literature for domestic animals, using horses as a study model (Table 2). For the four traits studied, few genomic windows explained most of the additive genetic variation, as illustrated in the Manhattan plots shown in Figure 2, Figure 3, Figure 4 and Figure 5.

For POS, three windows were found that explained more than 4% of the additive variance in the trait, in decreasing order: 51.31% (ECA17), 4.56% (ECA2), and 4.41% (ECA5) (Table 2). The sum of the variances explained by these windows was 60.28% of the total additive variance, resembling a qualitative trait, due to the large influence of few genomic regions on the trait.

Three windows that explained more than 4% of the additive genetic variance were also identified for NUM: 26.80% (ECA1), 6.46% (ECA19), and 4.25% (ECA23) (Table 2). The windows together explained 37.51% of the total genetic variance in the trait, following the pattern observed for POS.

The other two traits evaluated, NUML and NUMR, followed the same patterns of genetic architecture of the others: few genomic regions with a large influence on the trait. There were two windows explaining 14.06% (ECA5) and 8.48% (ECA11) of the additive genetic variance in NUML (sum of 22.54%), and only one window explaining 81.96% (ECA7) of the variance in NUMR (Table 2).

The genes were prospected according to their location in the genome, and arranged as shown in Table 3.

## 4. Discussion

This is the first GWAS of hair whorl traits in domestic animals, using horses as a study model. Hair whorl traits are used as indicators of behavior in horses [3,4,5,6]. The genetic architecture of these traits was very similar, with few genomic regions explaining much of the additive genetic variance in the traits. There were between one and three genomic windows that explained more than 4% of the additive variance for each trait. Together, these windows explained 20% to 80% of the additive variance in the traits analyzed. The main genomic windows associated with the different whorl phenotypes were not the same. However, the traits are highly correlated with one another [10]. This indicates that gene groups influence the traits to different extents. Similar results have been reported in the literature for other genetically correlated traits [13].

For POS, the window that explained most of the additive variance was located on ECA17. This window harbors the *KLF5* gene, which is known for its role in epidermal biology, as well as controlling repair and growth in the root of the hair follicle in humans [14]. This gene may therefore play a role in hair whorl formation. Curiously, this gene has been associated with chronic schizophrenia in humans [15]. This is an interesting fact, since hair whorl position on the head of animals, including horses, has been associated with behavior. This finding can be attributed to the pleiotropic effect of this major-effect gene. Another gene with an action similar to that of *KLF5* was found in the second window for POS, located on ECA2. The *IL2* gene is related to hair follicle growth, and has been associated with hair loss (alopecia) in humans [16]. In addition, an association with multiple sclerosis [17] and schizophrenia [18] has been reported, demonstrating its pleiotropic effect. 

Candidate genes found in the main windows associated with hair whorl position on the head of horses exert an effect on the formation and growth of hair follicles, and might be associated with whorl formation. Interestingly, the genes identified here have a pleiotropic effect on psychiatric characteristics. This is the first indication of biological and physiological activity that provides a plausible explanation for hair whorl position as an indicator of behavior. Furthermore, it is the beginning of the observation/validation of the genetic mechanisms underlying the phenotypic observation.

For NUM, the window explaining most of the additive variance, located on ECA1, contains the *SIRT1* gene, which is expressed in the hair follicle [19,20], and may participate in hair whorl formation. This gene has also been associated with depression and schizophrenia [21]. Other genes with an action similar to that of *SIRT1* were found in the second window for NUM. The *CD47* gene is known for its biological role in follicular development and hair follicle formation [22], in addition to activity in the central nervous system, participating in the cortical development of neurons [23] and regulating recovery processes in the central nervous system after severe injuries in humans [24]. The *CD200* gene is expressed in hair follicle stem cells [25], and is related to schizophrenia-like alterations in animals [26], in addition to Parkinson’s disease [27]. The third window for NUM, located on ECA23, contains the *ALDH1A1* gene, which is involved in signaling processes and morphogenesis of the hair follicle [28]. An association with Parkinson’s disease has also been reported [29]. 

Similar to the position trait, genes associated with the number of hair whorls on the head exert activity in both the hair follicle and neurodegenerative diseases, which are indicators of behavioral and neurological alterations. The number of hair whorls on the head has also been associated with behavior in horses [7], providing additional evidence of the interrelationship between these phenotypes.

Analysis of the main window associated with NUML, located on ECA5, revealed a series of genes expressed in the hair follicle: *NECTIN4* [30], *PPOX* [31], *CCDC190* [32], *NCSTN* [33], *PBX1* [34], *VANGL2* [35] and *KLHDC9* [32]. The number of hair whorls on the neck has not yet been associated with behavioral traits, but there is evidence of neurological action of some of the cited genes: *NCSTN* [36], *PBX1* [37,38], and *VANGL2* [39]. A series of genes expressed in the hair follicle were also found in the second window, located on ECA11: *TP53* [40], *AURKB* [41], *PER1* [42], *ALOXE3* and *ALOX12B* [43] and *DVL2* [44]. Some of these genes are associated with hair loss: *ALOX15B* [45], *SHBG* [46], and *ALOX15* [47]. In addition, there is evidence of neurological action of some of the genes identified in the second window: *PER1* [48], *SHBG* [49,50], and *DVL2* [51]. 

For NUMR, some genes related to hair follicle development and growth were found in the window that explained the highest proportion of variance, located on ECA7; these are listed as candidates: *RHOG* [52], *UCP2* [53], *UCP3* [54] and *IL18BP* [55]. Although this trait has not been associated with behavior, some of the genes found also exhibit neuropsychiatric actions: *RHOG* [56], *UCP2* [57], and *UCP3* [58].

Unfortunately, the low SNP density leads to large genomic windows with many genes. This makes it difficult to identify reliable candidates. The use of high-density panels would improve finding them and fine-map causative mutations. 

## 5. Conclusions

Genomic regions associated with hair whorl traits in horses were identified in the present study. In these regions, many genes whose metabolic activity is related to hair follicle growth were prospected, and they were indicated as candidates that may influence these traits. Curiously, some of these genes also have known neurological and behavioral functions. The possible pleiotropic effect of these genes is the first indication of genetic–biological validation and elucidation of the association between hair whorls and temperament in animals.

## Figures and Tables

**Figure 1 animals-11-02925-f001:**
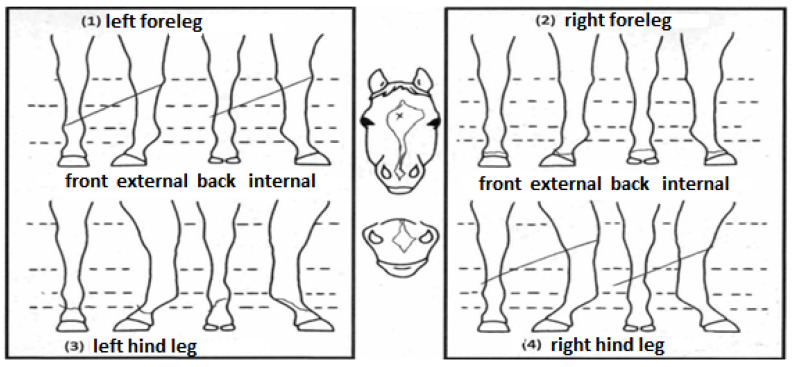
Official outline diagram from ABQM containing a graphic representation of the whorls’ position and number (authors’ own translation).

**Figure 2 animals-11-02925-f002:**
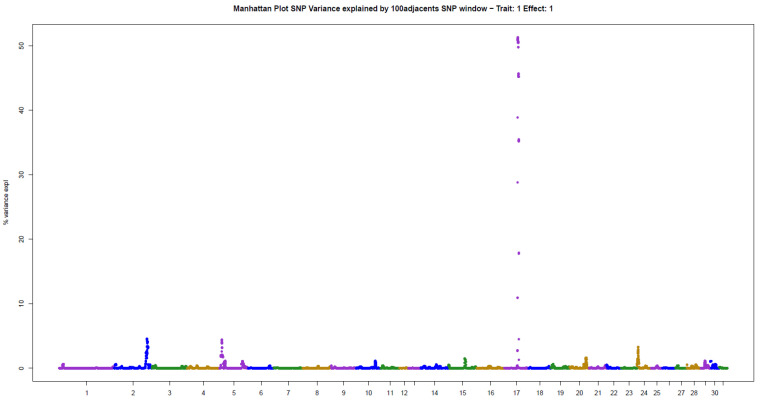
Manhattan plot of hair whorl position on the head of Quarter Horses. The y-axis indicates the proportion of variance explained by windows of 100 adjacent SNPs. The chromosomes are given on the x-axis.

**Figure 3 animals-11-02925-f003:**
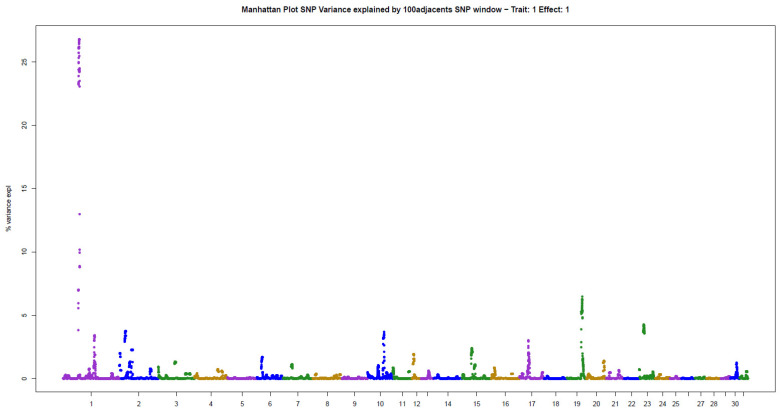
Manhattan plot of hair whorl number on the head of Quarter Horses. The y-axis indicates the proportion of variance explained by windows of 100 adjacent SNPs. The chromosomes are given on the x-axis.

**Figure 4 animals-11-02925-f004:**
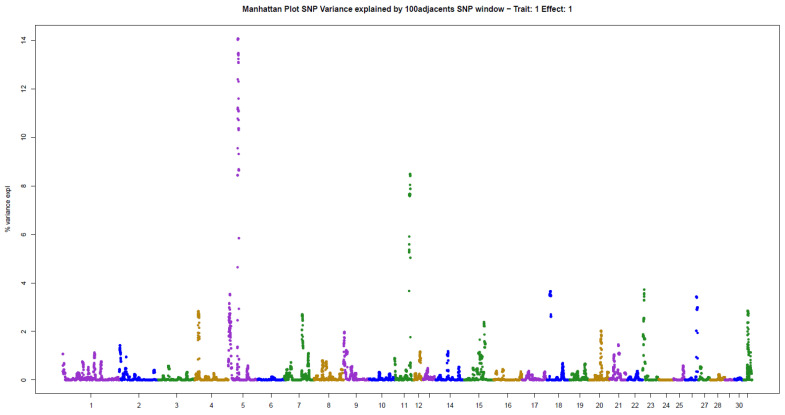
Manhattan plot of hair whorl number on the left side of the neck of Quarter Horses. The y-axis indicates the proportion of variance explained by windows of 100 adjacent SNPs. The chromosomes are given on the x-axis.

**Figure 5 animals-11-02925-f005:**
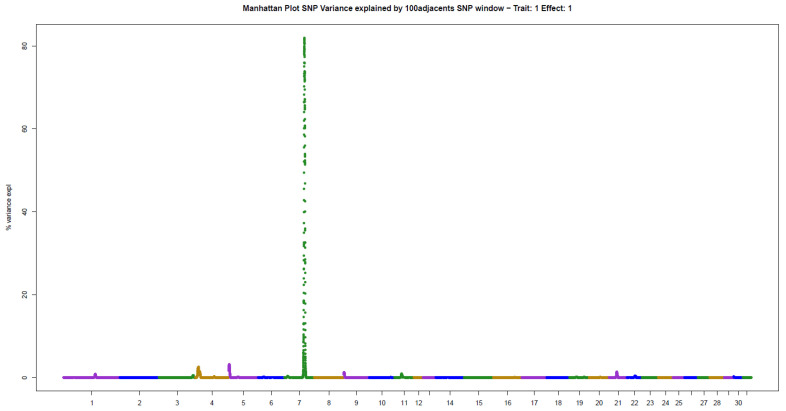
Manhattan plot of hair whorl number on the right side of the neck of Quarter Horses. The y-axis indicates the proportion of variance explained by windows of 100 adjacent SNPs. The chromosomes are given on the x-axis.

**Table 1 animals-11-02925-t001:** Descriptive statistics of hair whorl traits in Quarter Horses.

Trait	Phenotype (No. of Animals)	Total No. of Animals
POS *	1(161), 2(144), 3(18)	323
NUM **	1(306), 2(32), 3(2)	340
NUML **	1(263), 2(65), 3(10), 5(2)	340
NUMR **	1(263), 2(65), 3(11), 7(1)	340

POS: position of hair whorls on the head; NUM: number of hair whorls on the head; NUML and NUMR: number of hair whorls on the left and right side of the neck, respectively. * For hair whorl position on the head, the phenotypes were divided as follows: (1) above the upper eye line; (2) between the upper and lower eye line; and (3) below the lower eye line. ** The number of hair whorls were counted on the outline diagram to obtain their numerical values.

**Table 2 animals-11-02925-t002:** Genomic windows and percentage of additive genetic variance (Va%) in the four hair whorl phenotypes explained by each window.

Phenotype	Genomic Window (Chromosome: Start Position–End Position in bp)	Va%
POS	17: 45,252,386–49,471,423 2: 104,112,526–108,667,352 5: 7,191,873–11,827,369	51.31 4.56 4.41
NUM	1: 54,370,083–58,251,385 19: 49,710,771–53,302,322 23: 17,020,869–20,697,071	26.80 6.46 4.25
NUML	5: 29,306,930–34,034,348 11: 50,127,066–53,788,216	14.06 8.48
NUMR	7: 70,662,347–74,512,078	81.96

POS: position of hair whorls on the head; NUM: number of hair whorls on the head; NUML and NUMR: number of hair whorls on the left and right side of the neck, respectively.

**Table 3 animals-11-02925-t003:** Genes within significant windows for each trait.

Trait	Chromosome/Additive Variance Explained (%)	Gene (Symbol)
POS	ECA17 (51.31)	*KLF5, PIBF1, MZT1, KLF12, BORA, DACH1, LMO7, UCHL3, DIS3, TBC1D4, COMMD6*
	ECA2 (4.56)	*IL2, MIR147B, SMIM43, USP53, C2H4orf3, PDE5A, MAD2L1, PRDM5, NDNF, TNIP3, QRFPR, ANXA5, CCNA2, TRPC3, BBS12, NUDT6, EXOSC9, BBS7, KIAA1109, ADAD1, SPATA5, SPRY1, ANKRD50, FABP2, FGF2, IL21*
	ECA5 (4.41)	*FASLG, TNFSF4, MIR214, MIR199A, TNFSF18, TEX50, KIAA0040, RABGAP1L, COP1, TNR, TNN, CACYBP, SERPINC1, DARS2, KLHL20, ANKRD45, SLC9C2, SUCO, C5H1orf05, PIGC, DNM3, MYOC, FMO1, FMO2, MROH9, PAPPA2, MRPS14, RC3H1, CENPL, PRDX6, METTL13, VAMP4, PRRC2C, FMO4, FMO3 ZBTB37, GPR52*
NUM	ECA1 (26.80)	*SIRT1, VPS26A, HK1, SUPV3L1, DDX21, STOX1, TET1, SLC25A16, DNA2, MYPN, FAM241B, TSPAN15, SRGN, KIFBP, DDX50, CCAR1, HNRNPH3, RUFY2, PBLD, HERC4, DNAJC12, LRRTM3, CTNNA3, ATOH7, NEUROG3, HKDC1, TACR2*
	ECA19 (6.46)	*CD47, CD200, HHLA2, CCDC54, MYH15, TRAT1, BBX, IFT57, CIP2A, DZIP3, RETNLB, GUCA1C, MORC1, DPPA4, DPPA2*
	ECA23 (4.25)	*ALDH1A1, MIR204B-2, MIR8951, C23H9orf57, APBA1, PTAR1, CFAP95, MAMDC2, SMC5, CEMIP2, C23H9orf85, TRPM6, FAM189A2, KLF9, TRPM3, ABHD17B, GDA, ZFAND5, RORB, C23H9orf40, TMC1*
NUML	ECA5 (14.06)	*NECTIN4, PPOX, CCDC190, NCSTN, PBX1, VANGL2, KLHDC9, FCER1G, ADAMTS4, MIR7177A, ATF6, PCP4L1, SLAMF9, FCRLB, F11R, NIT1, TOMM40L, CFAP126, ARHGAP30, UFC1, USP21, B4GALT3, NDUFS2, APOA2, NR1I3, SDHC, RGS5, HSD17B7, CD244, LY9, SLAMF7, UHMK1, CD84, SPATA46, SLAMF6, PEX19, LMX1A, PEA15, NUF2, CASQ1, RGS4, ATP1A4, DDR2, ATP1A2, SH2D1B, IGSF8, KCNJ9, KCNJ10, PIGM, IGSF9, CFAP45, CD48, SLAMF1, COPA, DCAF8, TAGLN2, HSPA6, FCRLA, OLFML2B, DUSP12, MPZ, TSTD1, USF1, PFDN2, DEDD, NHLH1*
	ECA11 (8.48)	*TP53, AURKB, PER1, ALOXE3, ALOX12B, ALOX15, ALOX15B, SHBG, DVL2, MYH1, MYH2, SLC2A4, EFNB3, TNFSF13, MIR195, MIR497, MIR324, MIR9096, TNFSF12, SPEM2, TMEM220, RNASEK, FXR2, BCL6B, KCTD11, WRAP53, FGF11, PHF23, DNAH2, ADPRM, SCO1, GAS7, GLP2R, DHRS7C, USP43, STX8, NTN1, MFSD6L, KRBA2, ODF4, ARHGEF15, CTC1, VAMP2, GUCY2D, CHD3, KDM6B, ATP1B2, SAT2, POLR2A, SLC35G6, CHRNB1, SPEM1, NLGN2, PLSCR3, TMEM95, ACAP1, YBX2, CLDN7, ELP5, CTDNEP1, ASGR2, SLC16A11, SLC16A13, C11H17orf49, PELP1, PIRT, MYH3, MYH8, MYH13, CFAP52, PIK3R5, PIK3R6, CCDC42, MYH10, NDEL1, RNF222, RPL26, RANGRF, SLC25A35, PFAS, BORCS6, TMEM107, CNTROB, CYB5D1, NAA38, TMEM88, TMEM102, TMEM256, TNK1, NEURL4, GPS2, ACADVL, DLG4, ASGR1, GABARAP, GSG1L2, SENP3, EIF5A, RCVRN, SOX15, ZBTB4, MYH4, HES7, TRAPPC1, KCNAB3, MPDU1, EIF4A1, CD68*
NUMR	ECA7 (81.96)	*RHOG, UCP3, IL18BP, UCP2, SLCO2B1, MIR139, MIR326, COA4, ATG16L2, TRIM21, ART5, ANAPC15, FOLR1, FOLR2, INPPL1, CLPB, FCHSD2, P2RY2, ARHGEF17, RELT, PLEKHB1, RAB6A, MRPL48, C2CD3, PPME1, P4HA3, KCNE3, LIPT2, CHRDL2, RNF169, NEU3, ARRB1, KLHL35, GDPD5, SERPINH1, MAP6, MOGAT2, RRM1, STIM1, PGAP2, NUP98, ART1, NUMA1, LAMTOR1, FOLR3, PHOX2A, PDE2A, ARAP1, STARD10, FAM168A, PAAF1, PGM2L1, POLD3, SPCS2, RPS3, OR52B3, TPBGL, P2RY6, OR52B37P, OR52M2, OR52P2, OR51AE1, OR52B4OP, OR52B4F, OR52B4GP, OR52B4E, OR52B4, OR55B1, CHRNA10, RNF121, XRRA1, OR2AT2, OR2AT2D, OR2AT2EP, OR2AT13P, OR52K1, OR52B4N, OR52B4D, DNAJB13*

POS: position of hair whorls on the head; NUM: number of hair whorls on the head; NUML and NUMR: number of hair whorls on the left and right side of the neck, respectively.

## Data Availability

The data presented in this study are available on request from the Rogério Abdallah Curi.

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
