# Peer review of "Genomic Regions Associated with the Position and Number of Hair Whorls in Horses"

_animals, 2021, doi:10.3390/ani11102925_

Round 1
Reviewer 1 Report
The study is aimed at detection of genomic regions associated with hair whorls in the light of their relationship with behavioral traits of the horses.
In my opinion the study is original and well written. I cannot find major problems with this manuscript. I have only some minor comments:
- Please include PCA for your population under study to visualize its genetic differentiation and possible stratification
- Please add some information on SNP data filtering from previous study
- Please discuss the study limitations of which the major is relatively low SNPs density and high number of detected genes which hampers reliable identification of functional candidates.
- It is pity that there is no information on behavioral traits of the analyzed horses. This would supplement the study.
I recommend minor revision.
Author Response
Dear editor and reviewer,
We would like to thank all the corrections and suggestions. The comments improved the manuscript and highlighted the results presented. All the suggestions were accepted. An explanation was given to each comment of the reviewers using the acronym AU (author). Modifications were highlighted in yellow.
Reviewer 1
The study is aimed at detection of genomic regions associated with hair whorls in the light of their relationship with behavioral traits of the horses.
In my opinion the study is original and well written. I cannot find major problems with this manuscript. I have only some minor comments:
- Please include PCA for your population under study to visualize its genetic differentiation and possible stratification
AU: We calculated PC for genotypes. There is no stratification of the population as it can be seen in the Figure below. So, we did not consider the PCA in the analyses.
- Please add some information on SNP data filtering from previous study
AU: We added information about SNP data filtering (lines 102-107).
- Please discuss the study limitations of which the major is relatively low SNPs density and high number of detected genes which hampers reliable identification of functional candidates.
AU: We agree with the reviewer and the limitations were presented (lines 260-262).
It is pity that there is no information on behavioral traits of the analyzed horses. This would supplement the study.
AU: It is true. We are motivated by the results, and we are planning to continue the studies including behavioral traits.

Reviewer 2 Report
The manuscript is interesting and explores a GWAS of a novel trait and a of interest to the equine industry. My major concern is the statistical analysis. A threshold model was implemented given there are classes defined in NUM and NUML and NUMR with only 1 or 2 observations. Extremely low incidence in a class is problematic from a statistical point of view as it leads to a situation similar to the extreme case problem (ECP). Biologically, it makes little sense to consider an extra discrete class when the response is observed only in one or two individuals.
Comments:
Line 25. Not clear. Reword. “The traits exhibited one to three genomic windows that explained more than 4% of the additive variance”
Line 26: “20% to 80% of the additive variance of each analyzed trait.” So is it 4% per window for three windows as you mentioned earlier ?
Line 37: “ explores” change. The word does not fit.
Figure 1 Should be translated to English
Line 102: For clarification. Each trait was implemented separately and not a multi-trait model was used? Please clarify
Line 104: Since a threshold model was used and there are classes defined in NUM and NUML and NUMR with 1 or 2 observations. Extremely low incidence in a class is problematic from a statistical point of view as it leads to a situation similar to the extreme case problem (ECP). Biologically, it makes little sense to consider an extra discrete class when the response is observed only in one or two individuals.
Line 118-120: Some information on convergence should be provided.
Author Response
The manuscript is interesting and explores a GWAS of a novel trait and a of interest to the equine industry. My major concern is the statistical analysis. A threshold model was implemented given there are classes defined in NUM and NUML and NUMR with only 1 or 2 observations. Extremely low incidence in a class is problematic from a statistical point of view as it leads to a situation similar to the extreme case problem (ECP). Biologically, it makes little sense to consider an extra discrete class when the response is observed only in one or two individuals.
AU: We totally agree with the reviewer. In order to lead with the situation, we grouped the classes with few observations to another ones with more observations and re-ran the analyses. For the present dataset, the results did not change. So, we decide to keep with the analyses as they were presented firstly. Moreover, we indicated that the classes with few observation were grouped to other classes, the analyses re-ran and the results were the same (lines 95-97).
Comments:
Line 25. Not clear. Reword. “The traits exhibited one to three genomic windows that explained more than 4% of the additive variance”
AU: The sentence was rephrased for better understanding (lines 25-26).
Line 26: “20% to 80% of the additive variance of each analyzed trait.” So is it 4% per window for three windows as you mentioned earlier ?
AU: The sentence was rephrased for better understanding (lines 25-26).
Line 37: “ explores” change. The word does not fit.
AU: The sentence was corrected (line 37).
Figure 1 Should be translated to English
AU: Figure was translated (lines 71-72).
Line 102: For clarification. Each trait was implemented separately and not a multi-trait model was used? Please clarify
AU: Traits were implemented separately. A better clarification was provided (lines 112-114).
Line 104: Since a threshold model was used and there are classes defined in NUM and NUML and NUMR with 1 or 2 observations. Extremely low incidence in a class is problematic from a statistical point of view as it leads to a situation similar to the extreme case problem (ECP). Biologically, it makes little sense to consider an extra discrete class when the response is observed only in one or two individuals.
AU: An explanation was given above (first question).
Line 118-120: Some information on convergence should be provided.
AU: Convergence information was provided (lines 124-125).
